# Supportive Care in Oncology—From Physical Activity to Nutrition

**DOI:** 10.3390/nu14061149

**Published:** 2022-03-09

**Authors:** Thorsten Schmidt, Philip Süß, Dominik M. Schulte, Anne Letsch, Wiebke Jensen

**Affiliations:** 1University Cancer Center Schleswig-Holstein, University Hospital Schleswig-Holstein, Campus Kiel, 24105 Kiel, Germany; anne.letsch@uksh.de; 2Association for Health and Rehabilitation Sports at UKSH, 24105 Kiel, Germany; 3Institute of Diabetes and Clinical Metabolic Research, University Hospital Schleswig-Holstein, Campus Kiel, 24105 Kiel, Germany; philip.suess@uksh.de (P.S.); dominik.schulte@uksh.de (D.M.S.); 4Division of Endocrinology, Diabetes and Clinical Nutrition, Department of Internal Medicine 1, University Hospital Schleswig-Holstein, Campus Kiel, 24105 Kiel, Germany; 5Department of Oncology and Hematology, BMT with Section Pneumology, Hubertus Wald Tumor Center—University Cancer Center Hamburg, University Medical Center Hamburg-Eppendorf, 20246 Hamburg, Germany; w.jensen@uke.de

**Keywords:** physical activity, exercise, nutrition, supportive therapy, oncology

## Abstract

The diagnosis and treatment of cancer are associated with impairment at the physical and at psychological level. In addition, side effects are a potentially treatment-limiting factor that may necessitate dose reduction, delay, or even discontinuation of therapy, with negative consequences for outcome and mean survival. Numerous studies have shown that physical activity and sports and exercise therapy programs are not only practicable but also recommendable for oncologic patients during the acute phase and in the aftercare. Furthermore, nutrition plays an important role in all stages of tumor therapy. A timely integration of a nutrition therapy and physical activity in the form of physiotherapy and sports therapy serves to prevent and reduce treatment-associated side effects. Evidence-based recommendations on cancer prevention through nutrition therapy, physical activity, and sports and exercise therapy should be integrated into treatment plans for oncology patients as well as in health care services for the general population. Individual counselling by trained nutrition and exercise specialists may be advisable to receive concrete recommendations on the respective tumor entity or specific side effects. This mini review is based on a selective literature search in the PubMed database and Cochrane Central Register of Controlled Trials on the subjects of healthy diet and physical activity in primary prevention and follow-up about cancer.

## 1. Introduction

Cancer incidence has risen worldwide over the last years. A cancer diagnosis is associated with medical therapies, such as surgery, chemotherapy, or radiation, with physiologic effects [1]. Various studies describe the safety of physical activity during therapy and in the aftercare. Based on the growing numbers of studies, the evidence about the influence of physical activity and exercise therapy, healthy weight, and nutrition therapy in cancer treatment rises [2,3,4]. 

The selective literature search in the PubMed database and Cochrane Central Register of Controlled Trials included randomized controlled trials (RCT), review articles, and meta-analyses with the subjects of healthy diet and physical activity in primary prevention and follow-up about cancer. A search was conducted in January 2022. 

## 2. Physical Activity and Exercise Therapy in Different Therapy Phases

In the past, clinicians advised cancer patients to rest and avoid physical activity. Physical activity and exercise therapy with cancer patients were not performed to the same degree as, e.g., for coronary heart disease. The rehabilitation of oncological patients has focused on reducing the side effects of surgical therapy, such as pain, and on reducing general restrictions, and there were no specific recommendations for physical activity and exercise therapy in the various treatment phases. Inactivity can weaken the skeleton, cause muscle loss, and lead to fat gain, with increases in the risk of a negative outcome in relation to body composition, obesity-related diseases, frailty and fractures, and cancer recurrence. Exercise research in the last decades is available in which the supportive care outcomes and the impact of physical activity and exercise therapy are described. There is evidence that the short and long adverse effects of medical treatment might be prevented, reduced, or treated through physical activity and exercise therapy. Despite the good study results and the available evidence on the subject of “physical activity in oncology”, only every second patient took part in sports and exercise therapy for oncological patients during medical therapy or was not informed about the options for sports and exercise therapy [5]. 

### 2.1. Physical Activity and Exercise Therapy during Chemotherapy and Radiotherapy 

In a randomized and controlled study, 230 breast cancer patients undergoing chemotherapy received either low-intensity home-based training, moderate to strenuous supervised endurance and strength training, or no intervention in the control group. After six months of follow-up, the activity groups showed a reduced decline in cardiopulmonary performance and physical functions as well as less pain, nausea, and vomiting compared to the control group [6]. Similar results were reported in an RCT. The results highlighted the improvements in strength, endurance, and quality of life from the exercise training in comparison to the control group [7]. Further positive study results on physical activity parallel to adjuvant therapy are available, for example, with prostate cancer or breast cancer patients during radiotherapy [8,9]. Studies on the use of physical activity as part of inpatient treatment with non-small-cell lung cancer patients, with patients after a hematological stem cell transplant, and with lymphoma patients report a stabilization of exercise intensity and walking distance, an improvement in quality of life, and a reduction in fatigue symptoms [10,11,12]. Overall, it was found that the dose of physical activity is the decisive parameter in order to achieve the positive effects of physical activity. While nausea, tiredness, and general weakness predominate under chemotherapy treatment, the most common permanent side effect by far is chemotherapy-induced peripheral polyneuropathy (CIPNP) [13,14]. Chemotherapeutic agents containing platinum drugs (e.g., cisplatin), taxanes (e.g., paclitaxel), or vinca alcaloids (e.g., vinblastine and vincristine) are particularly associated with a high risk of CIPNP. Several studies from the field of sports and exercise medicine show promising success in the treatment of CIPNP with regular guided sensorimotor and vibration plate training [15,16,17]. In an RCT with breast cancer patients taking paclitaxel, sensorimotor training was able to achieve significantly reduced fluctuations in the monopedal position [16]. Similar results are available for lymphoma patients. A training intervention led to an improvement in depth sensitivity in 87.5% of the cases. Furthermore, the patients reported an improvement in their subjectively perceived quality of life [18]. Furthermore, studies have shown that the severity of lymphedema in breast cancer patients can be reduced by strength training or upper body ergometer training. This is based on the activation of the muscle-vein pump and, in parallel to a compression stocking, causes a reduction in lymphedema through increased lymphatic and venous return with less flow into the interstitium [19,20].

### 2.2. Physical Activity and Sports and Exercise Therapy in the Aftercare

The diagnosis and the medical treatment of cancer are associated with a severe reduction in quality of life, which can persist even after treatment. After the surgical intervention and the side effects of other therapies, the patients are often in a phase of fear and uncertainty, both towards their own bodies and towards their families and their surroundings. One goal of rehabilitation is to alleviate these limitations and speed up recovery after therapy. In addition to psychological therapy, sports and exercise therapy are just as important building blocks in rehabilitation. The side effects of the therapy, such as fatigue, nausea, and reduced muscular and cardiovascular performance, can be reduced through exercise training [21]. 

Whereas, in the past, the goal of sports and exercise therapy in tertiary prevention or rehabilitation was initially only seen as improving quality of life and fatigue, epidemiological observational studies also show a decrease in the recurrence rate with greater physical activity compared to inactive people [22]. The American Cancer Society Nutrition and Physical Activity Guidelines for Cancer Survivors recommended a healthy lifestyle including healthy body weight, physical activity, and a diet that includes vegetables, fruits, and whole grains. Following the recommendation was associated with a longer survival after the diagnosis of stage III colon cancer [23]. Similar results were observed in the Nurse Health Study with 2987 stage I-III breast cancer patients, with a risk reduction through physical activity ranging from 26% in the less active group to 40% in the most active group [24].

### 2.3. Physical Activity and Sports and Exercise Therapy in Palliative Care 

Due to the development of new chemotherapies and modern targeted therapy strategies, the survival of patients with advanced metastatic cancers has been extended considerably in many tumor entities. Nevertheless, persistent disease-related symptoms and the additional therapy-related side effects significantly limit the patients’ quality of life [25]. The value of physical activity and exercise therapy in advanced cancer patients is still unclear. Nevertheless, in patients with advanced cancer, exercise has the potential to prevent the loss of function, control symptoms, and help maintain independence [26]. Recent reviews have evaluated that exercise interventions are safe and feasible in advanced cancer patients and in those with metastatic bone disease [27,28,29,30]. Furthermore, recent reviews and meta-analyses also showed that exercise interventions achieve potential benefits on quality of life, fatigue, cancer-related symptoms, and functional status outcomes [31,32]. Prospective controlled trials with larger sample sizes are needed to adequately analyze the efficacy of exercise programs in patients with advanced cancer. 

### 2.4. Type, Dose, and Timing of Activity

The specific exercise recommendations are described in the “Exercise Guidelines for Cancer Survivors” [2]. In the context of sports and exercise therapy, all main motor skills should have a different focus in the training schedule, depending on the entity, stage of the disease, the therapy phase, and the side effects. The content, the methods, and the progression of the training are based on the individual abilities and goals of the patient. Depending on the points mentioned, the training schedule is designed individually so that the focus of two cancer patients with the same cancer disease can be different due to different treatment phases or side effects. 

In addition to the individual physical strain, the degree of supervision seems to be decisive for the success of a physical activity. A review article from 2017 reports that a guided intervention for physical activity is superior to self-guided activity in terms of improving quality of life and body awareness [33].

### 2.5. FITT-Criteria

The American College of Sports Medicine (ACSM) Roundtable is a multidisciplinary group of international experts that first published evidence reviews on exercise and cancer and related exercise recommendations for cancer survivors in 2010 and updated them in 2019 [2,34,35]. Systematic reviews, meta-analyses, and randomized controlled trials on the topic of exercise therapy in cancer were evaluated by the panel. On this basis, the evidence of the studies was assessed for targeted disease- and therapy-related symptoms [2]. Furthermore, targeted exercise recommendations, so-called FITT criteria (frequency, intensity, time, and type), for aerobic and/or resistance training were formulated. The group concluded that there is strong evidence regarding the beneficial effects of targeted exercise therapy on anxiety, depression, fatigue, health-related quality of life, and physical functioning. Moderate evidence was formulated for bone health and sleep quality. Exercise studies on cardiotoxicity, chemotherapy-induced peripheral polyneuropathy, cognitive function, falls, nausea, pain, sexual function, and treatment tolerability, among others, were assessed with weak evidence [2]. 

In addition, a general exercise recommendation for cancer patients has been evaluated, which includes moderate-intensity aerobic training at least 3 times per week, for at least 30 min, for at least 8 to 12 weeks. The addition of resistance training to aerobic training, at least 2 times per week, using at least 2 sets of 8 to 15 repetitions of at least 60% of the 1-repetition maximum, appears to result in similar benefits [2]. With this evidence, assessment, and the formulation of the FITT criteria, the recommendations of the ACSM 2019 for cancer survivors have been significantly specified compared with the results of 2010.

### 2.6. Exercise Testing and Contraindications 

To create an individualized exercise prescription, cancer survivors should receive a comprehensive assessment of all components of health-related physical fitness (i.e., cardiorespiratory fitness, muscular strength and endurance, body composition, and flexibility). Campbell et al. (2019) also formulated additional cancer-specific considerations for a comprehensive assessment. However, it is not always possible to perform a comprehensive physical fitness assessment before starting exercise training. For this reason, most survivors do not require testing to begin low-intensity aerobic exercise (e.g., walking or cycling), resistance training with gradual increases, or a flexibility program [2]. Furthermore, exercise prescription should be individualized to the cancer survivor’s prior treatment, his or her aerobic fitness, response to treatment, concomitant medical conditions, and the immediate or persistent adverse effects of treatment occurring at any given time [35]. Schmitz et al. (2010) also provide an overview of the objectives for exercise in cancer survivors as well as general and cancer-specific contraindications to starting an exercise program, reasons for stopping exercise, and injury risk guidelines [35]. 

## 3. Nutrition and Nutrition Assessment

Nutrition Care is not only important in the prevention of cancer but plays a role in cancer treatment compatibility. The nutritional counselling of oncology patients is complex and can include the goals of reducing and treating malnutrition, improving treatment outcomes, and improving quality of life and must be continually reviewed during therapy.

### 3.1. Nutrition Assessment

Starting with each nutritional intervention, a comprehensive nutritional assessment should first be carried out. To measure body composition, the non-invasive measurement method of multifrequency bioimpedance analysis in both outpatient and inpatient settings has proven to be very practicable. In order to assess the state of health before, during, and after the intervention, special attention should be paid to the parameters of phase angle, body cell mass, and the ratio between extracellular mass/body Cell Mass (ECM/BCM). In addition, retrospective food diaries should be collected by the nutritionist in order to detect dietary errors at an early stage and, thus, be able to correct them in a targeted manner. In addition, patients should receive regular care to implement the recommendations in everyday life. A detailed social and medical anamnesis should be an integral part of the assessment in order to be able to classify the individual goals of the patients in the overall context and, consequently, to be incorporated into the course of therapy planning. A detailed nutritional assessment is, thus, the basis for an individualized nutritional therapy, which in turn is irreplaceable to avoid the risk of malnutrition.

### 3.2. Nutrition Problems

Even before diagnosis, many patients experience unwanted weight loss, which suggests possible malnutrition in the majority of patients with cancer. This weight loss is caused by insufficient energy intake, whereby, in particular, too little protein intake favors the breakdown of body cell mass and the catabolic metabolic situation [36]. Since side effects such as emesis, nausea, pain, and gastrointestinal disorders can lead to a reduced food intake and, at the same time, to a reduction in macro- and micronutrient intake, nutritional therapy should, consequently, be integrated into the overall therapy from the time of diagnosis. The primary goal of nutritional therapy is, thus, the avoidance of malnutrition, whereby the individual therapy-related side effects should serve as a guide for the contents of the nutritional intervention. In particular, the preservation of the fat-free mass should be aimed at in cooperation with the movement therapists [37].

### 3.3. Calorie Intake

Basically, the daily energy supply of the patients should be through protein, fat, and carbohydrates. To measure individual calorie requirements, indirect calorimetry has developed into the gold standard [38]. In practice, the rule of thumb of 25–30 kcal/kg body weight has proven itself [4], but there are also differences here with regard to the determined and the recommended energy requirement [37]. A generalization of energy requirements can, therefore, lead to a reduction in fat-free mass and malnutrition in the long term. Regular documentation of body weight and daily calorie intake is, therefore, recommended, as even a small daily deficit can have a lasting negative influence on the course of therapy [39]. Although there are already results regarding possible positive effects of controlled energy restriction through continuous or intermittent fasting, these findings are not meaningful enough to draw clear conclusions [40]. It should also be considered that calorie restriction can increasingly lead to muscle and fat loss and, thus, contribute to cachexia-related sarcopenia [40]. Stabilization of body weight can, thus, improve the course of therapy and prognosis of the patient in the long term [41,42,43].

### 3.4. Macronutrients

In addition to meeting energy requirements, special attention should be paid to the daily protein intake [44]. To compensate for the altered protein metabolism, the recommendation is to maintain a daily protein intake of 1.2–1.5 g/kg bodyweight/day [3,40] so that excessive loss of muscle mass can be avoided. Elderly patients should increase their protein intake daily to 1.5 g/kg bodyweight, or about 15–20 percent of energy, to prevent the loss of muscle mass and to optimize muscle function [45]. In the case of already missing muscle mass, however, it is not yet known whether 1.5 g/kg bodyweight protein is sufficient to favorably influence the body composition. Therefore, it is obvious that the previous recommendations of 1.2–1.5 g/kg bodyweight need to be updated or at least adapted based on the patient’s body composition, as previous weight loss is associated with a loss of muscle mass and, thus, influences protein requirements. In addition to the total protein intake, the time of protein intake and the amino acid composition of the food should be considered. In particular, the branched-chain amino acids are of particular importance here since leucine and its derivatives can improve protein synthesis [37,46]. Patients should be encouraged to maintain an even daily protein intake, with additional carbohydrate- and protein-rich post-workout meals [37]. Such large amounts of protein with high proportions of branched-chain amino acids can be achieved by small meals with a variety of milk and dairy products. Fat-reduced dairy products are particularly suitable for dysphagic patients or patients with nausea. In addition, dairy and dairy products are suitable for patients with inflammation in the mouth and throat and can be enriched very well with carbohydrate or protein products. These products can also be combined very well with high-quality fatty acids and, thus, provide patients with kilocalories and anti-inflammatory foods. As is well known, the use of high-quality vegetable oils can have a positive effect on inflammation. Consequently, the use of omega-3 fatty acids, monounsaturated fatty acids, and the reduction of arachidonic and trans fatty acids should be taken into account in nutritional therapy [38]. A high intake of omega-3 fatty acids can, therefore, not only reduce the incidence of cancer but also reduce cancer-associated symptoms [47,48,49]. A reduction in high-fat animal products as well as highly processed foods, thus, reduces the intake of omega-6 and trans fatty acids, whereby the inflammatory process can be positively influenced [50]. However, this should not result in a low-fat diet, but only increase the proportion of foods with monounsaturated fatty acids and omega-3 fatty acids. In patients with weight loss and insulin resistance, a change in the fat–carbohydrate ratio should be considered to reduce the risk of infection from hyperglycaemia [51]. In addition, a carbohydrate-modified diet in favor of protein and fat intake generally appears to be more digestible for patients. In addition, there are contradictory but promising study results regarding a ketogenic “cancer diet” [52,53]. However, due to a lack of study data, the implementation of a ketogenic or low-carbohydrate diet cannot currently be recommended [54].

### 3.5. Micronutrients and Supplements

In addition to a demand-covering supply of macronutrients, an equivalent level of attention should be paid to the daily coverage of micronutrients. Already in the last decade, as well as increasingly in the pandemic period, the interest in health-promoting nutrient preparations has grown, which makes this topic no less relevant for oncological patients. Although studies with potential benefits already exist regarding individual micronutrients and trace elements, these results are not yet meaningful enough for a recommendation [42]. It is recommended that vitamin and trace element products should be administered only if there is an indication of a deficiency condition. For all micronutrients and trace elements, the recommendations of the national societies and the guideline of the European Society for Clinical Nutrition and Metabolism (ESPEN) [42,55] apply to substitution. The supplementation of nutrients should only be carried out in consultation with the attending physician. 

### 3.6. Dietary Regimens

For some years now, various cancer diets have been propagated, which are associated with promises to favorably influence the course of therapy and disease. One of the well-known diets is the ketogenic diet. The low-carbohydrate or ketogenic diet is presented both as an alternative healing method and as a basis for the effectiveness of chemotherapy or radiotherapy. It is also presented as an accompanying measure to improve the tolerability of this therapy. To date, no scientific evidence has been provided for this form of diet with positive results in terms of survival, an improvement in therapy response, or therapy tolerance. In contrast, clinical data on deficiency symptoms can be found so there is currently no indication for a cancer diet [56].

Several studies have shown that adherence to the Mediterranean diet can reduce the risk of cardiovascular disease [57] and inflammation [58] as well as mortality due to breast cancer [59]. The Mediterranean diet describes a nutritional concept that is based on vegetables, grains, and nuts and is characterized by a proportionally high intake of olive oil and fish, which are rich in the n-3 and n-9 polyunsaturated fatty acids (PUFA), as well as a moderate intake of dairy products and low consumption of red meat, which contain primarily n-6 PUFA [60].

### 3.7. Practical Implementations

The focus of nutritional intervention should be on the avoidance of malnutrition, whereby, in particular, the coverage of energy and protein requirements should be strictly monitored and adhered to. An adjustment of carbohydrate intake or an increase in fat intake should be made after a thorough assessment of nutritional behavior and should always be individually adapted to nutritional needs. An anti-inflammatory diet with a targeted intake of omega-3 fatty acids and a simultaneous reduction of omega-6 and trans fatty acids also appears to be very promising. In addition, daily intakes of fruits, vegetables, and legumes [61] should be met for micronutrient coverage, eliminating the need for the supplementation of nutrients above the recommended intakes. In this regard, the western diet, with a variety of highly processed foods and a high intake of omega-6 fatty acids from high-fat animal foods and sunflower, safflower, and corn oil, appears less favorable than a Mediterranean diet [37,62]. In particular, the high intake of olive oil [63] seems to have positive effects on the prevention of some cancers and on the prevention of noncommunicable diseases [37,44]. It is important to consider that the Mediterranean diet generally has a more favorable fatty acid profile and a better ratio of omega-6 to omega-3 fatty acids due to a lower consumption of high-fat animal foods and a higher proportion of maritime foods. Thus, the beneficial effects of the Mediterranean diet could be understood as the sum of dietary habits and food choices rather than the result of a single food. In general, a plant-based and meat-reduced diet seems to have positive effects on health [49], which automatically makes a vegan or vegetarian diet the first choice for many patients. The disadvantage of these plant-based diets may be the daily coverage of the increased protein requirement since the low protein density of plant foods may result in a high consumption of those. The high intake of plant-based dietary fibers can, thus, lead to a feeling of satiety at an early stage or to gastrointestinal complaints. The supposedly correct approach of this diet can, thus, lead to a protein deficiency in some cases. Due to the lack of fatty sea fish, additional consideration should be given to covering omega-3 fatty acids with algae oils. Thus, a plant-based diet can be considered for the treatment of patients with cancer if protein coverage, the fatty acid profile, and critical nutrient coverage are in accordance with the indication-related recommendations [37]. Due to the therapy-related complications during radio- and chemotherapy, it is important to perform a nutritional assessment including a measurement of body composition at an early stage to specifically identify nutritional problems and to be able to correct them at an early stage. The focus should not only be on food selection, meal frequency, and dietary habits but also, in particular, on the preparation of meals. Within the framework of a standardized process model, such as the NCP or GNCP, the nutritionist should permanently re-evaluate the recommendations based on a renewed assessment and classify them in the context of a multimodal therapy.

## 4. Conclusions and Future Directions

Supportive care in oncology comprises optional methods that, if well-coordinated with the actual tumor therapy, can be used primarily with the aim of improving the quality of life, promoting patient empowerment, and reducing side effects. Nutrition and physical activity are two relevant supporting components in the treatment of oncological diseases and should be an integral part of an oncological therapy concept. International standards and scientific evidence show that exercise and nutrition therapy can prevent a wide range of problems associated with the disease and therapy. Specific exercise recommendations for fatigue, lymphedema, and improvement of quality of life and physical function are described in the ACSM Exercise Guidelines for Cancer Survivors [2]. 

Substitution of, e.g., micronutrients and trace elements should only be conducted in consultation with the medical team. The guidelines of the European Society for Clinical Nutrition and Metabolism (ESPEN) are as follows. The implementation of a ketogenic or low-carbohydrate diet cannot currently be recommended. In order to inform patients in a timely and comprehensive manner, the topic of nutrition should be actively addressed by physicians and healthcare professionals.

An important fact to consider in all possible preventive and supportive mechanisms of physical activity and nutritional therapy is continuity. Short-term interventions are unlikely to prevent the process of tumor development over months to years, while regular physical activity and nutritional therapy can lead to various adaptive mechanisms with improved defenses against cancer. 

Despite a large body of supportive evidence, sports and exercise therapy and nutrition therapy are not included in the standard treatment plans of cancer patients. For a successful implementation of sports and exercise therapy and nutritional intervention in oncology treatment, interdisciplinary exchange is essential. 

## Data Availability

Not applicable.

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
