# Peer review of "Supportive Care in Oncology—From Physical Activity to Nutrition"

_nutrients, 2022, doi:10.3390/nu14061149_

Round 1

Reviewer 1 Report

In my opinion, the submitted manuscript meets aims and scope of „Nutrients” Journal and Special Issue „Nutrition and Physical Exercise in the Patients with Cancer” and may be accepted after revision.

  1. In my opinion, submitted manuscript is a review, no original research manuscript, so the authors should correctly mark the type of publication (line 1).
  2. The citation in the line 40, could be written as [2-4], not [2][3,4].
  3. According to the PRISMA guidelines (http://prisma-statement.org/PRISMAStatement/Checklist.aspx) authors should: Specify the inclusion and exclusion criteria for the review and how studies were grouped for the syntheses. Specify all databases, registers, websites, organisations, reference lists and other sources searched or consulted to identify studies. Present the full search strategies for all databases, registers and websites, including any filters and limits used. Specify the methods used to decide whether a study met the inclusion criteria of the review, including how many reviewers screened each record and each report retrieved, whether they worked independently, and if applicable, details of automation tools used in the process… The manuscript does not describe how the literature for this review was collected. Only in the abstract  it is mentioned, that the PubMed database was searched (line 31) – please complete this.
  4. I think, that in the sentence: „There is evidence, that…(line 52)”, the world „effects” is missing. Do you mean the short and long adverse effects of the cancer therapy?
  5. The abbreviations „RCT” (randomized clinical trial?) and PDCAAS should be explained, when they first appear in the text (line 63 and 246).
  6. In my opinion, the sentence: „Chemotherapeutic agents containing…” (line 75) should be rewritten as: „Chemotherapeutic agents containing platinum drugs (e.g. cisplatin), taxanes (e.g. paclitaxel) or vinca alcaloids (e.g. vinblastine, vincristine) are particularly associated with a high risk of CIPNP.”
  7. The lines 109-111 ought to be deleted („This section may be…”).
  8. The citation in the line 227 is [4,4] – is it a random/accidental repetition?
  9. In my opinion the Conclusions section is too general. I would appreciate if authors could give a very brief summary of their conclusions here, and be more specific.

Author Response

Thank you for the review. As an appendix you will find a point by point answer.

Reviewer 2 Report

- First of all, this manuscript can not be considered as a research article but strictly a narrative minireview, within, the background is relatively narrow and still poorly organized, the authors stated that
their review is based on a selective literature search in the PubMed database, therefore authors must
underlined these selective criteria and there restrictions,
- The format of the review for the journal Nutrients must also be respected, for example :
Introduction, Relevant Sections, Discussion, Conclusions, and Future Direction,
- The introduction is very short and must be detailed in such a way that it reflects the content of the
review,
-The content although fair and well written must be expanded, more details are needed.
- What would be the added value of this work compared to existing reviews, the authors must
explicitly demonstrate that in the abstract, in the whole text as well as in the conclusion?
- in paragraph 2.1, other data are needed with more references, the ideal would be to group the
results in the form of a table, mention the type of cancer, side effects, type of the study if systematic
reviews, meta-analyses, and randomized controlled trials…., the years of the study, the population
studied, the conclusions significance etc …
- furthermore, it is well recognized that chronic undernutrition, overnutrition as well as physical
exercise have an important roles in oncology, however more comprehensive assessment is required,
in this sense, have you excluded references that did not provide any recommendations more
particularly for specific nutrition in oncology during treatment of cancer, in the aftercare and even
in palliative care ?
- authors have also to discuss in another paragraph some dietary regimens including western, vegan
eastern, mediteranean among many under exploration, their beneficial effects and their limitations
in oncology
minor points
- add a comma in line 98 : ‘’Whereas in the past,’’
- from line 109 to 111 : what section are you talking about ?, rephrase or remove,
‘’This section may be divided by subheadings. It should provide a concise and precise description
of the experimental results, their interpretation, as well as the experimental 110 conclusions that can
be drawn. ‘’
- add references in paragraph 3.2 line 218-220 ‘’Although there are already results regarding
possible positive effects of 218 controlled energy restriction through continuous or intermittent
fasting, these findings are not meaningful enough to draw clear conclusions ‘’
- there are no abbreviations list…..for example for PDCAAS (Protein Digestibility Corrected
Amino Acid Score)
correct ref. 16

Author Response

Thank you for the review. We have marked the manuscript as a review and described the literature search.
Since in the reviews various wishes no change requests were made to the structure or , unfortunately not all change requests could be implemented. We apologize for this. 

Round 2

Reviewer 2 Report

please I need point by point answers on all the comments I already made in the first report, otherwise it will be hard for me to decide

Author Response

Dear ladies and gentleman, 

thank you for your expert optinion. 

Point by point answers

First of all, this manuscript can not be considered as a research article but strictly a narrative minireview,

Answer: We have made the correction and identified the publication as a review

within, the background is relatively narrow and still poorly organized, the authors stated that
their review is based on a selective literature search in the PubMed database, therefore authors must underlined these selective criteria and there restrictions,

Answer: We have extended the search string described

- The format of the review for the journal Nutrients must also be respected, for example :
Introduction, Relevant Sections, Discussion, Conclusions, and Future Direction,

Answer: We have added the points Conclusions, and Future Direction. A discussion of the individual points takes place in the sub-items.

We were approached last year by the Guest Editor for a manuscript on Nutrition and Physical Exercise in the Patients with Cancer. When asked, we were told that a general article was requested - with no specific intentions (review, RCT). Therefore, the other structure was created.

- The introduction is very short and must be detailed in such a way that it reflects the content of the
review,
-The content although fair and well written must be expanded, more details are needed.
- What would be the added value of this work compared to existing reviews, the authors must
explicitly demonstrate that in the abstract, in the whole text as well as in the conclusion?
- in paragraph 2.1, other data are needed with more references, the ideal would be to group the
results in the form of a table, mention the type of cancer, side effects, type of the study if systematic
reviews, meta-analyses, and randomized controlled trials…., the years of the study, the population
studied, the conclusions significance etc …

Answer: In point 2.1, the effects of physical activity under therapy are described and substantiated with the current literature. We cannot see the added value of adding further sources, as the relevant sources are mentioned.

- furthermore, it is well recognized that chronic undernutrition, overnutrition as well as physical
exercise have an important roles in oncology, however more comprehensive assessment is required,
in this sense, have you excluded references that did not provide any recommendations more
particularly for specific nutrition in oncology during treatment of cancer, in the aftercare and even
in palliative care ?

Answer: We have rewritten the point 3.6 in which recommendations are made

- authors have also to discuss in another paragraph some dietary regimens including western, vegan
eastern, mediteranean among many under exploration, their beneficial effects and their limitations
in oncology minor points

Answer: We have extended the manuscript with the paragraph diets (ketogenic diet and mediterranean diet)

- add a comma in line 98 : ‘’Whereas in the past,’’

Answer: Done

- from line 109 to 111 : what section are you talking about ?, rephrase or remove,
‘’This section may be divided by subheadings. It should provide a concise and precise description
of the experimental results, their interpretation, as well as the experimental 110 conclusions that can
be drawn. ‘’
- add references in paragraph 3.2 line 218-220 ‘’Although there are already results regarding
possible positive effects of 218 controlled energy restriction through continuous or intermittent
fasting, these findings are not meaningful enough to draw clear conclusions ‘’
- there are no abbreviations list…..for example for PDCAAS (Protein Digestibility Corrected
Amino Acid Score)
correct ref. 16

Answer: Done

With kind regards, 

Thorsten Schmidt